# TRIM25 promotes the cell survival and growth of hepatocellular carcinoma through targeting Keap1-Nrf2 pathway

Yanfeng Liu [1,2,5], Shishi Tao[1,5], Lijuan Liao[1], Yang Li[1], Hongchang Li[1], Zhihuan Li[3], Lilong Lin[3], Xiaochun Wan[1], Xiaolu Yang[4]* & Liang Chen [1]*

Tumor cells often exhibit augmented capacity to maintain endoplasmic reticulum (ER) homeostasis under adverse conditions, yet the underlying mechanisms are not well defined. Here, through the evaluation of all human TRIM proteins, we find that TRIM25 is significantly induced upon ER stress. Upregulation of TRIM25 ameliorates oxidative stress, promotes ER-associated degradation (ERAD), and reduces IRE1 signaling in the UPR pathway. In contrast, depletion of TRIM25 leads to ER stress and attenuates tumor cell growth in vitro and in vivo. Mechanistically, TRIM25 directly targets Keap1 by ubiquitination and degradation. This leads to Nrf2 activation, which bolsters anti-oxidant defense and cell survival. TRIM25 expression is positively associated with Nrf2 expression and negatively with Keap1 expression in hepatocellular carcinoma (HCC) xenografts and specimens. Moreover, high TRIM25 expression correlates with poor patient survival in HCC. These findings reveal TRIM25 as a regulator of ER homeostasis and a potential target for tumor therapy.

[1] Shenzhen Laboratory of Tumor Cell Biology, Institute of Biomedicine and Biotechnology, Shenzhen Institutes of Advanced Technology, Chinese Academy of Sciences, Shenzhen 518055, P. R. China. [2] Stem Cell Research Center, Ren Ji Hospital, School of Medicine, Shanghai Jiao Tong University, Shanghai 200127, China. [3] Dongguan Enlife Stem Cell Biotechnology Institute, Zheshang Building, #430 Dongguan Ave., Dongguan, Guangdong 523000, China. [4] Department of Cancer Biology and Abramson Family Cancer Research Institute, Perelman School of Medicine, University of Pennsylvania, Philadelphia, PA 19104, USA. [5] These authors contributed equally: Yanfeng Liu, Shishi Tao. *email: xyang@pennmedicine.upenn.edu; liang.chen@siat.ac.cn

Solid tumor cells are frequently exposed to various intrinsic and microenvironmental perturbations that trigger adaptive responses to favor cancer cell survival and progression[1]. The endoplasmic reticulum (ER) is fundamental for protein biosynthesis, modifications and trafficking. Protein homeostasis in the ER is therefore extremely sensitive to certain stimuli regarding oncogenic activation, hypoxia and nutrient deprivation, which often disrupts ER functions, leading to the accumulation of improperly folded proteins, an event known as ER stress[2]. To cope with this, tumor cells have evolved integrated signaling networks to facilitate the protein folding and elimination capacity. Endoplasmic reticulum (ER)-associated degradation (ERAD) and the unfolded protein response (UPR) are two key quality-control machineries in the cell[3]. ERAD is responsible for the clearance of misfolded proteins in the ER, while UPR is activated in response to ER stress that controls expression of many ERAD genes through three branches of the UPR signaling initiated by protein kinase RNA (PKR)-like ER kinase (PERK), activating transcription factor 6 alpha (ATF6α), and inositol-requiring kinase 1 (IRE1)[4–6].

The UPR represents an adapting response to restore ER homeostasis and promote tumor cell survival. However, chronic ER stress activates mechanisms that result in cell death. Aberrant UPR signaling was correlated with enhanced tumor cell growth, invasion capacity, and was associated with worse clinical outcome in multiple cancer types[7–9]. Recently, genome-wide sequencing revealed somatic alterations in genes encoding UPR sensors. Particularly, somatic mutations of PERK, IRE1, and ATF6 were found in several cancers, supporting a potential clinical relevance of UPR signaling with tumor progression[10–13].

The Keap1-Nrf2 pathway is well-established as the master signaling cascade that governs cellular defense against oxidative stresses. Under normal conditions, Keap1 binds to and sequesters Nrf2 in the cytoplasm, resulting in proteasomal degradation[14,15]. Following oxidative stress, Nrf2 is released from Keap1, translocates into the nucleus where it activates ARE-mediated detoxifying enzyme gene expression including HO1 and NQO1[16]. Aberrant Nrf2 signaling was frequently found in multiple cancers including HCC, which was linked to tumorigenesis and tumor progression. However, the underlying mechanisms regarding to the regulation of Nrf2 signaling have not been fully elucidated.

Tripartite motif-containing (TRIM) family members commonly contain a conserved RING domain at the N-terminus, most of which process E3 ubiquitin ligase activities. Accumulating evidence has shown that TRIM family proteins play significant roles in many physiological disorders including innate immunity, tumorigenesis and neurodegenerative disease, predominantly by governing the protein quality control[17–20]. We and others have previously reported that TRIM19 removes nuclear misfolded proteins by sequential SUMOylation and ubiquitination, and TRIM11 effectively eliminates both nuclear and cytosolic misfolded proteins, and promotes tumorigenesis[20–22]. The human genome is predicated to encode more than 70 members of TRIM proteins, the functions of TRIM proteins in regulating ER protein homeostasis remained largely unknown. Therefore, it is of great interest to explore the potential TRIM family members that might be involved in the modulation of ER stress.

In this study, we perform a systematic interrogation of gene expression encoding TRIM family members during ER stress, and identify TRIM25 as the most significantly induced gene in response to ER stress. As a feedback mechanism, TRIM25 is required for tumor cells to defend against ER stress. Functional investigation reveals that TRIM25 facilitates tumor cell survival by targeting Keap1 for ubiquitination and degradation, leading to activate Nrf2 signaling and reduce ROS levels during ER stress in

several cellular models of cancers. Furthermore, the expressions of TRIM25 are markedly upregulated in multiple cancerous tissues, and increased TRIM25 expression is associated with worse clinical outcome. Therefore, our study highlights a crucial regulator implicated ER stress, and suggests a potential therapeutic target for cancer intervention.

## Results

**TRIM25 functions as a feedback mechanism that responds to ER stress in tumor cells.** To evaluate the role of TRIM family members in the ER stress, we examined the expression levels of all TRIM genes (TRIM1-74) in HCT116 colon cancer cells following treatment with ER stress inducing drugs, TM (Tunicamycin) or TG (Thapsigargin). The BiP/GRP78 gene, one of ER stress markers[23], was used as the positive control. Results of real-time PCR showed BiP was significantly increased (Fig. 1a), suggesting the successful induction of ER stress. Concurrently, some TRIM genes were significantly upregulated, such as TRIM2, TRIM25, TRIM48, and TRIM49, while TRIM19 and TRIM31 were found to be decreased. No obvious changes were observed among other TRIM genes (Fig. 1a), indicating TRIM member proteins displayed gene-specific modulation of ER stress. Given its top enrichment, TRIM25 was selected to further investigate the functional link with ER stress. Western blotting verified that the protein level of TRIM25 was indeed increased upon treatment with either TM or TG in HCT116 and Huh7 cell lines, which is similar with GRP78, a master regulator of ER stress (Fig. 1b). Furthermore, TRIM25 ablation significantly upregulated the mRNA level of GRP78 (Fig. 1c, Supplementary Fig. 1a), suggesting that TRIM25 is required for ER homeostasis.

Considering that GRP78 is a master regulator for ER stress that activates URP signaling, leading to the upregulation of a broad UPR downstream genes, we then determined which target gene(s) would be responsive to TRIM25-mediated UPR inactivation. The mRNA level of sXBP1, which is linked to the IRE1 signaling pathways, was significantly up- or downregulated when TRIM25 is either depleted or overexpressed in the HCT116 and MCF7 cells (Fig. 1d, e, and Supplementary Fig. 1b, c). While ATF4 and CHOP, the downstream effectors of PERK signaling cascade, were only slightly altered (Fig. 1d, e, and Supplementary Fig. 1b, c), suggesting TRIM25 primarily suppresses IRE1 signaling pathways during ER stress. Accordingly, the phosphorylation of JNK but not eIF2α, two downstream effectors for IRE1 and PERK separately[2], was remarkably increased upon TRIM25 knockdown in the HCT116 cells (Fig. 1f). On the other hand, exogenous expression of TRIM25 downregulated the protein level of phospho-JNK andphospho-eIF2α upon ER stress in the HCT116 cells (Fig. 1g). Consistently, a similar effect was observed in a variety of tumor cells including human hepatocellular carcinoma Huh7, breast cancer MCF7 and osteosarcoma U2OS cells (Fig. 1h, i, and Supplementary Fig. 1d–f), indicating a common mechanism for TRIM25 in response to ER stress in various tumors. Taken together, these findings indicate that TRIM25 is a crucial regulator upon ER stress and negatively controls UPR signaling pathway.

**TRIM25 promotes ERAD activity by controlling ROS production.** Crosstalk between ERAD and UPR pathways serves as key quality-control machineries for the maintenance of ER homeostasis. To reveal the roles of TRIM25 in ER stress, we determined whether TRIM25 contributes to the clearance of misfolded proteins through ERAD. By evaluating the protein stability of CD3-δ-YFP, an ER marker and also a classical ERAD substrate[24], we found TRIM25 co-localized with CD3-δ-YFP and knockdown of TRIM25 showed increased protein abundance of

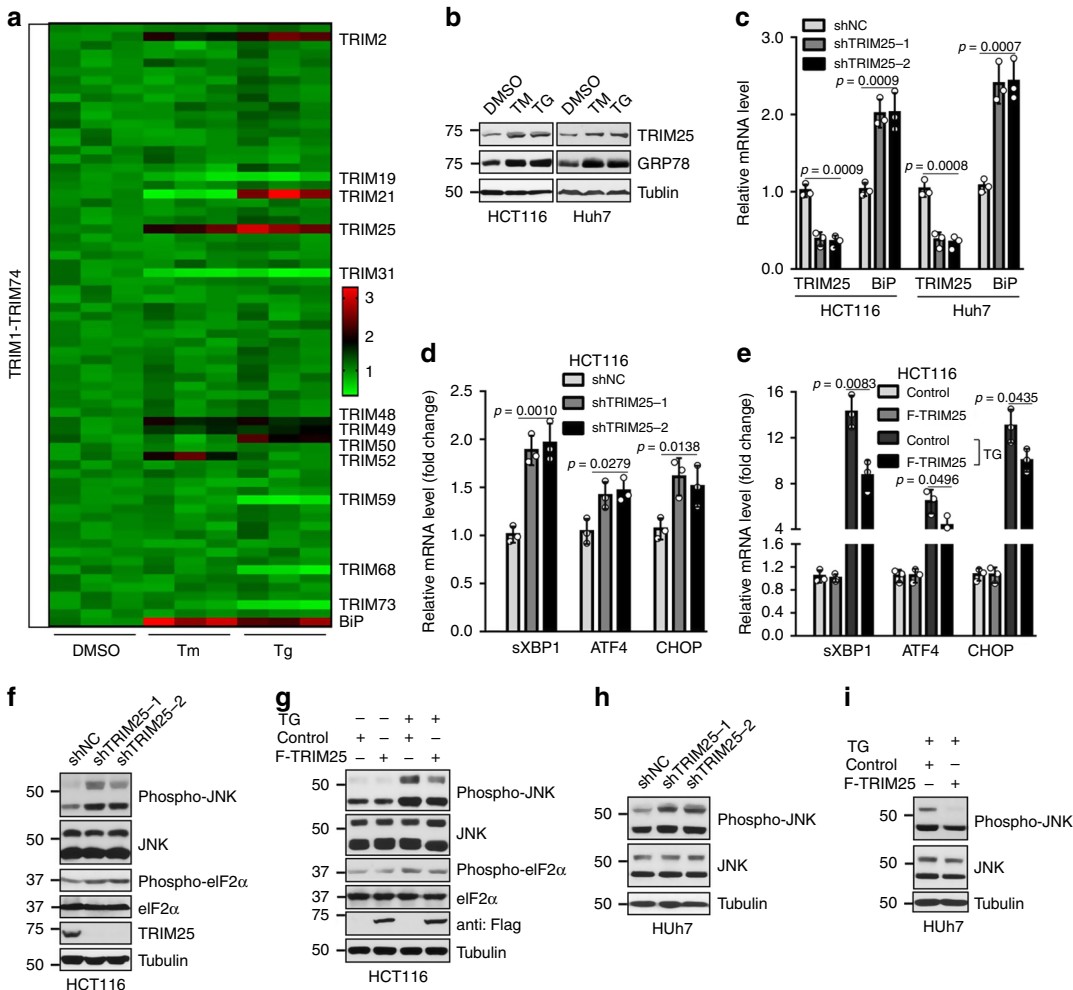

**Fig. 1 TRIM25 functions as a feedback mechanism that responds to ER stress in tumor cells. a** Relative mRNA fold change of all TRIMs in HCT116 cells treated with TM (Tunicamycin, 5 μg/ml) or TG (Thapsigargin, 1 μM) for 6 h. **b** Western blot analysis the level of TRIM25 and GRP78 in HCT116 and Huh7 cells treated with TM (5 μg/ml) or TG (1 μM) for 12 h. **c** Relative mRNA fold change of TRIM25 and BiP genes in HCT116 and Huh7 cells after stable knockdown of TRIM25. **d** Relative mRNA fold change of UPR genes: sXBP1, ATF4, CHOP in HCT116 cells with control (shNC) or stable knockdown of TRIM25. **e** Relative mRNA fold change of UPR genes: sXBP1, ATF4, CHOP in HCT116 cells stably overexpressing control or F-TRIM25 (Flag-TRIM25), treated with or without TG (1 μM) for 6 h. **f** Western blot analysis of UPR signaling, including phospho-JNK/JNK and phospho-eIF2α/eIF2α in HCT116 cells after stable knockdown of TRIM25 in HCT116 cells. **g** Western blot analysis of the levels of phospho-JNK/JNK andphospho-eIF2α/eIF2α in HCT116 cells overexpressing control or F-TRIM25, treated with or without TG (1 μM) for 12 h. **h, i** Western blot analysis of the levels of phospho-JNK/JNK in Huh7 cells after stable knockdown (**h**) or overexpression (**i**) of TRIM25, treated with or without TG (1 μM) for 12 h. For **c**–**e**, data represent the mean ± SEM (n = 3). Statistical significance was assessed using two-tailed Student's t-tests.

CD3-δ-YFP, as well as a marked decrease in its degradation (Fig. 2a–c, Supplementary Fig. 2a), suggesting impaired ERAD activity in the absence of TRIM25. Interestingly, TRIM25 facilitates CD3-δ-YFP degradation during ER stress by TG treatment (Fig. 2d, e). Given that accumulating evidence indicates altered endogenous ROS production is sufficient to disrupt protein folding and cause ER stress[25], we speculate that TRIM25 may modulate cellular ROS production in tumor cells. As expected, depletion of TRIM25 significantly increased ROS levels in the HCT116 and Huh7 cells (Fig. 2f–h and Supplementary Fig. 2b–e). Conversely, forced expression of TRIM25 did not affect ROS levels in resting tumor cells (Supplementary Fig. 2f, g). However, ROS production was significantly attenuated during ER stress in the presence of TRIM25 (Fig. 2i–l and Supplementary Fig. 2h). To examine the connection between ERAD and ROS, we treated TRIM25 depleted HCT116 cells expressing CD3-δ-YFP with the ROS scavenger N-acetyl-L-cysteine (NAC) or Nrf2 activator tert-butylhydroquinone (tBHQ). This partially rescued the decreased

ERAD, suggesting that TRIM25 positively regulating ERAD by partially controlling the cellular ROS level (Supplementary Fig. 2i). Taken together, these data here suggest that TRIM25 governs redox balance to maintain ER homeostasis.

**TRIM25 directly targets Keap1 to promote its ubiquitination and degradation.** To provide insight into the underlying mechanism how TRIM25 prevents tumor cells from ER stress in the certain microenvironment, co-immunoprecipitation (Co-IP) and mass spectrometry were conducted to identify potential substrates of TRIM25 related to ER stress. Keap1, an adapter protein targeting Nrf2 for ubiquitination and degradation, was identified as a potential interacting partner (Fig. 3a, b). Co-IP assays showed that TRIM25 indeed interacts with Keap1, and its interaction was significantly increased upon ER stress condition (Fig. 3c, d and Supplementary Fig. 3a). Similarly, GST pulldown assays indicated that TRIM25 could interact with Keap1 (Fig. 3e).

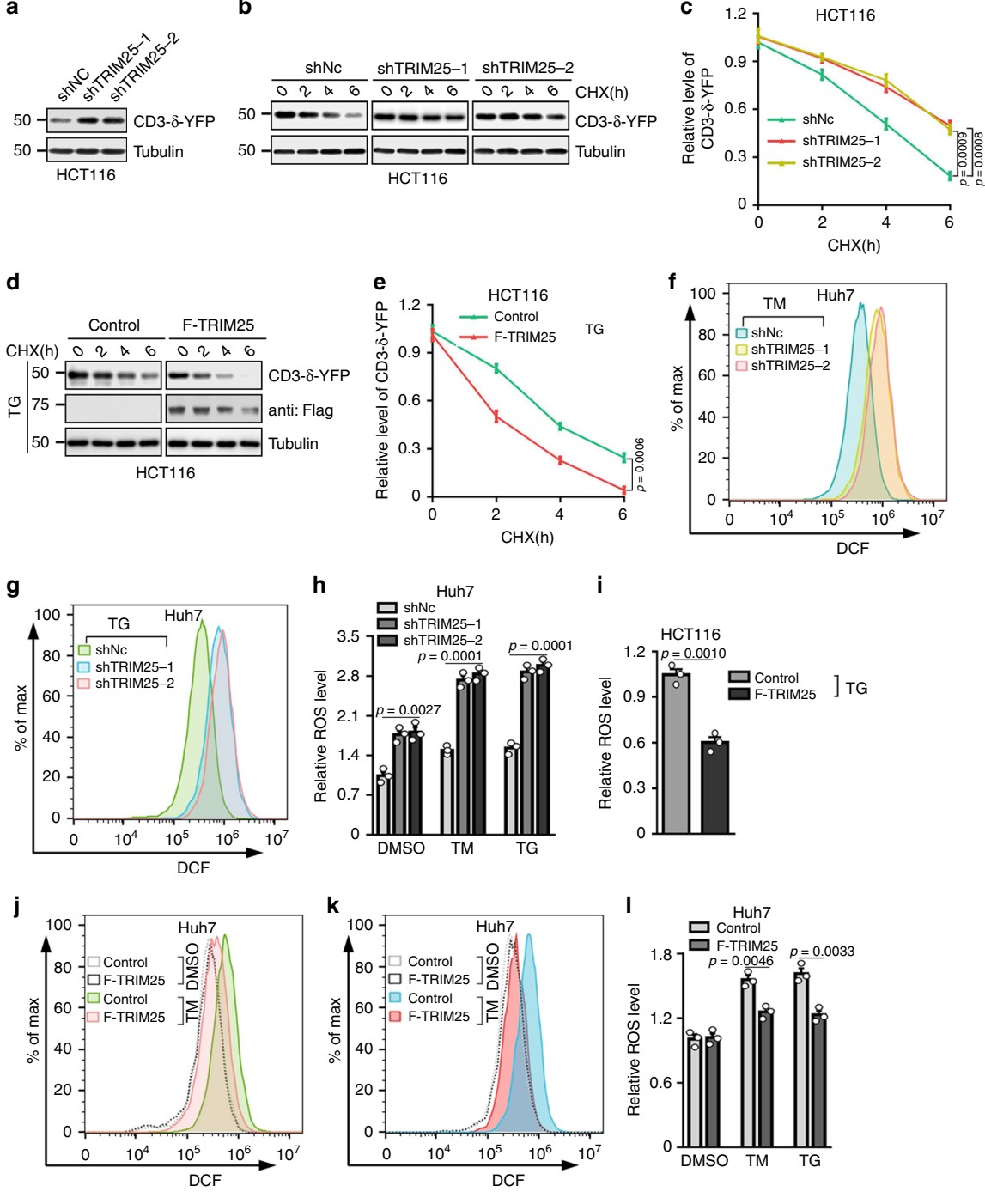

**Fig. 2 TRIM25 facilitates ERAD and reduces ER stress induced ROS levels in tumor cells. a** Western blot analysis of the levels of CD3-δ-YFP in HCT116 cells with control (shNC) or stable knockdown of TRIM25. **b, c** Half-life of CD3-δ-YFP in HCT116 cells with control or stable knockdown of TRIM25, treated with cycloheximide (CHX) at the indicated times and analyzed by western blot. Representative western blot (**b**) and quantified graph (**c**) are shown. In **b**, the exposures of the corresponding control and TRIM25-knockdown blots were adjusted to achieve comparable levels of CD3-δ-YFP at time 0. **d, e** Half-life of CD3-δ-YFP in control and F-TRIM25-expressing HCT116 cells pre-treated with TG (1 μM) for 6 h, then treated with cycloheximide (CHX) at the indicated times and analyzed by western blot. Representative western blot (**d**) and quantified graph (**e**) are shown. In **d**, the exposures of the corresponding control and F-TRIM25-expressing blots were adjusted to achieve comparable levels of CD3-δ-YFP at time 0. **f–h** Flow cytometry analysis of ROS levels in control and F-TRIM25-expressing Huh7 cells treated with TM (5 μg/ml) (**f**) or TG (1 μM) (**g**) for 6 h, and quantified data is shown as (**h**). **i** Relative ROS levels in control and F-TRIM25-expressing HCT116 cells treated with TG (1 μM) for 6 h. **j–l** Flow cytometry analysis of ROS levels in Huh7 cells for control and stable knockdown of TRIM25 treated with TM (5 μg/ml) (**j**) or TG (1 μM) (**k**) for 6 h, quantified data is shown as (**l**). For **c, e, h, i** and **l**, data represent the mean ± SEM (n = 3). Statistical significance was assessed using two-tailed Student's t-tests.

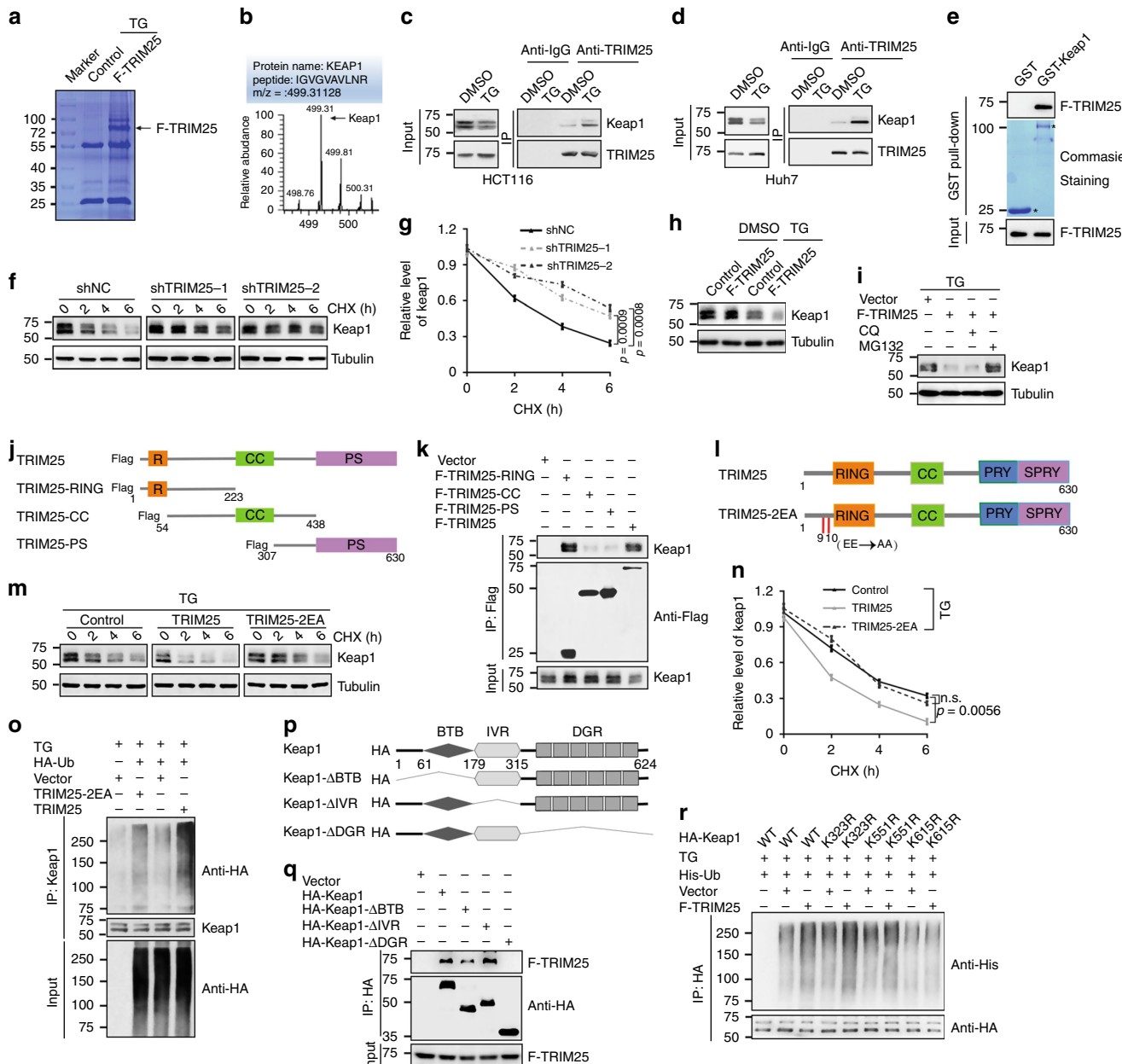

**Fig. 3 TRIM25 interacts and ubiquitinates Keap1. a** Identification potential substrates of TRIM25 related to ER stress by liquid chromatography-tandem mass spectrometry. **b** The Keap1 peptides identified through mass spectrometry are shown. **c**, **d** Co-immunoprecipitation (co-IP) assay analyzes the interaction of endogenous TRIM25 and Keap1 in HCT116 (**c**) and Huh7 (**d**) cells, treated with or without TG (1 μM) for 12 h. **e** In vitro GST pulldown assay analysis of the interaction of TRIM25 (F-TRIM25). Asterisks indicate the coomassie blue staining of GST and GST-Keap1. **f**, **g** HCT116 cells infected with shRNA lentivirus as indicated with treatment of TG (1 μM) for 6 h and then treated with CHX. Representative western blot (**f**) and the corresponding quantified graph (**g**) are shown. **h**, **i** Western blot analysis the level of Keap1 in HCT116 cells treated with or without TG (1 μM) for 12 h (**h**), and the above cells treated with TG (1 μM) simultaneously with MG132 (4 μM) or CQ (50 μM) for 12 h (**i**). **j** The RING zinc-finger (R), B-Box, coiled-coil (CC) and PRY/SPRY (PS) domain of TRIM25 are indicated. **k** Huh7 cells overexpressing full-length and TRIM25 truncates were immunoprecipitated with the indicated antibody. **l** TRIM25 and its mutant (TRIM25-2EA), the conserved Glu9 and Glu10 were mutated to Ala. **m**, **n** HCT116 cells overexpressing the indicated constructs with treatment of TG (1 μM) for 6 h. Representative western blot (**m**) and the corresponding quantified graph (**n**) are shown. **o** HCT116 cells transfected with the plasmids as indicated and treated with TG (1 μM) for 12 h. Immunoblot analysis of the cell lysates and Keap1-IP with the indicated antibodies. **p** The broad-complex, tramtrack and bric a brac (BTB) domain, intervening region (IVR) and the double glycine repeat (DGR) or Kelch repeat of KEAP1 were indicated. **q**, **r** Huh7 cells were transfected with the indicated plasmids. Immunoblot analysis of the HA-IP and cell lysates (**q**), and the ubiquitination of wide-type and Keap1mutaions with the indicated antibodies (**r**). For **g** and **n**, data represent the mean ± SEM (n = 3). Statistical significance was assessed using two-tailed Student's t-tests. n.s. not significant.

In addition, we found that knockdown of TRIM25 leads to a marked decrease in Keap1 degradation (Fig. 3f, g). Intriguingly, TRIM25 did not obviously alter the level of Keap1 under normal conditions, but noticeably reduced its protein level under ER stress (Fig. 3h). Moreover, this reduced Keap1 protein level can be rescued by proteasome inhibitor Z-Leu-Leu-Leu-al (MG132), but not autophagy inhibitor chloroquine (CQ) (Fig. 3i). Together, these results indicate that TRIM25 probably mediates Keap1 degradation by ubiquitin-proteasome pathway upon ER stress.

Next, we generated a series of truncated mutants of TRIM25 and Keap1 to dissect functional domains of the TRIM25-Keap1 interaction. For TRIM25, we constructed three TRIM25 truncates mutants, containing RING, coiled-coil (CC) and PRY/SPRY (PS) domain (Fig. 3j). And the immunoprecipitation assay showed that only TRIM25 N-terminal mutants containing RING domain interacted with Keap1 (Fig. 3k). As a ubiquitin E3 ligase, TRIM25 probably binds to and mediates the ubiquitination of Keap1, resulting in Keap1 degradation. To test this hypothesis, we constructed a TRIM25 mutant with Glu9 and Glu10 changed into Ala (termed as TRIM25-2EA) (Fig. 3l), which lost its ubiquitination activity[26]. Biochemistry data showed that exogenous expression the wild type (WT) of TRIM25, other than TRIM25-2EA significantly shortened the half-life of Keap1 during ER stress (Fig. 3m, n). In lines with this, enhanced ubiquitination level of Keap1 was found in Huh7 cells with the expression of WT TRIM25 instead of TRIM25-2EA mutant (Fig. 3o), indicating reduced protein stability of Keap1. Moreover, we mapped the interaction domains between TRIM25 and Keap1 mutants, and found that the double glycine repeat (DGR) domain of Keap1 was crucial for its interaction with TRIM25 (Fig. 3p, q). Furthermore, we constructed three Keap1 mutants with Lys323, Lys551, and Lys615 of DGR domain changed into Arg (termed as K323R, K551R, and K615R), and biochemistry data showed that an obvious suppression of the ubiquitination level of Keap1 was found when expression of Keap1-K615R mutant instead of other Lys mutants (Fig. 3r), suggesting TRIM25 directly targets K615 of Keap1 to promote its ubiquitination and degradation. To conclude, these results demonstrate that TRIM25 destabilizes Keap1 via enhancing its ubiquitination and degradation.

**TRIM25 activates Nrf2 signaling pathway through enhancing its nuclear translocation during ER stress.** As Keap1 is a well-known inhibitor of Nrf2[27], we next investigate whether TRIM25 regulates Nrf2 nuclear translocation and transcriptional activity due to TRIM25-dependent degradation of Keap1. Expectedly, TRIM25 in the HCT116 cells did not obviously affect the level of Nrf2 in both the cytoplasm and nucleus under basal condition (Fig. S3b), but robustly induced Nrf2 accumulation in the nucleus upon ER stress in the HCT116 and Huh7 cells (Fig. 4a, b). On the other hand, the protein levels of Nrf2 in nuclear extracts were reduced in HCT116 and Huh7 cells with TRIM25-knockdown under ER stress (Fig. 4c, d). Additionally, immunofluorescence (IF) analysis revealed TRIM25 markedly stimulates the nuclear translocation of Nrf2 in the U2OS cells under ER stress (Fig. 4e, f). Consistent with its altered localization, HO1 and NQO1, the direct targets of Nrf2, were significantly increased upon TRIM25 overexpression in the HCT116 and Huh7 cells specifically in response to ER stress (Fig. 4g, h), whereas depletion of TRIM25 noticeably attenuated their expressions (Fig. 4i, j). We then set to verify whether these observations are linked to the E3 ligase activity of TRIM25. In contrast to the effects of WT TRIM25, the TRIM25-2EA mutant failed to induce the nuclear translocation, transcriptional activity of Nrf2 (Fig. S3c and S3d), suggesting the E3 ligase activity is indispensable for TRIM25 to activate

Nrf2 signaling. Taken together, these findings show that TRIM25 activates Nrf2 signaling pathway through enhancing its nuclear translocation during ER stress.

**Nrf2 is required for TRIM25 to facilitate tumor cell survival during ER stress.** Considering that TRIM25 deficiency in tumor cells leads to ER stress, which compromises tumor survival, we therefore determine whether TRIM25 is involved in the survival and adaptation of tumor cells to ER stress. TRIM25 knockdown significantly decreased the cell viability in the Huh7 and HCT116 cells (Supplementary Fig. 4a, b). In addition, depleting TRIM25 induced the expression of activated caspase-3 (cleaved caspase3) in HCT116 and Huh7 cells (Fig. 5a, b), suggesting the activation pro-apoptotic program, which was further supported by flow cytometry analysis showing remarkably enhanced apoptosis in the HCT116 and Huh7 cells with TRIM25 ablation compared with control group (Fig. 5c, d, Supplementary Fig. 4c, d). These results reveal that the functionality of TRIM25 is required for ER homeostasis and tumor cell survival. Consistently, in the absence of ER stress, overexpression of TIRM25 did not elicit apparent effect on tumor cell viability in the HCT116 cells (Supplementary Fig. 4e). However, TRIM25 markedly rescued the growth suppression as well as the pro-apoptotic phenotype of tumor cells induced by ER stress in the HCT116 and Huh7 cells (Fig. 5e–h, Supplementary Fig. 4e, f). A similar effect was observed in MCF7 cells (Supplementary Fig. 4g, h). Moreover, we demonstrated the observation was dependent on the E3 ligase activity of TRIM25, as the TRIM25-2EA dead mutant was unable to reverse the ER stress induced apoptosis (Supplementary Fig. 4i–k).

Since we have demonstrated TRIM25 controls ER homeostasis and activates Nrf2 signaling cascade, it is of great importance to figure out whether TRIM25 indeed corporates with Nrf2 in response to ER stress. To test this hypothesis, we generated cell lines stably expressing TRIM25 in Huh7 and HCT116 simultaneously with Nrf2 depletion (Fig. 5i and Supplementary Fig. 5a). Nrf2 depletion almost abolished the functions of TRIM25 in the suppression of ROS production (Fig. 5j, k), inhibition of the UPR signaling pathways and promotion of ERAD during ER stress (Supplementary Fig. 5b–e). Moreover, the protecting role of TRIM25 for cell survival was significantly impaired in these cells with Nrf2 ablation (Fig. 5j, l, m). Altogether, the roles of TRIM25 that prevent ER stress and apoptosis are dependent on Nrf2.

**Knockdown of TRIM25 in HCC suppresses tumor growth in nude mice.** To examine the role of TRIM25 in vivo, we explanted Huh7 cells with control or TRIM25 knockdown in nude mice. Compared to the control, the TRIM25 knockdown group displayed significantly smaller tumors in nude mice (Fig. 6a–c). Furthermore, mice with TRIM25 knockdown have longer overall survival compared with the control group mice (Fig. 6d). In contrast to the control tumor xenografts of Huh7 origin, a reduction in Ki67 and Nrf2 expression, and upregulation of Keap1 expression were observed in the tumor xenografts derived from Huh7 cells with TRIM25 depleted (Fig. 6e, f), suggesting that TRIM25-induced Keap1-Nrf2 pathway plays a vital role in HCC growth. For its clinical significance in cancers, we then detected the expression levels of TRIM25, Keap1 and Nrf2 in cancerous and paired adjacent tissues from 90 HCC patients. A strong staining for Keap1 was observed in 22.2% (10/45) of the TRIM25 High expression group in comparison to 66.7% (30/45) of the TRIM25 Low expression group, and a strong Nrf2 staining in nuclear was observed in 73.3% (33/45) of the TRIM25 High group compared with 37.8% (17/45) in the TRIM25 low group (Fig. 6g, h). In addition, the expression level of TRIM25 and Nrf2 were also inversely correlated to Keap1 in breast tumor tissues

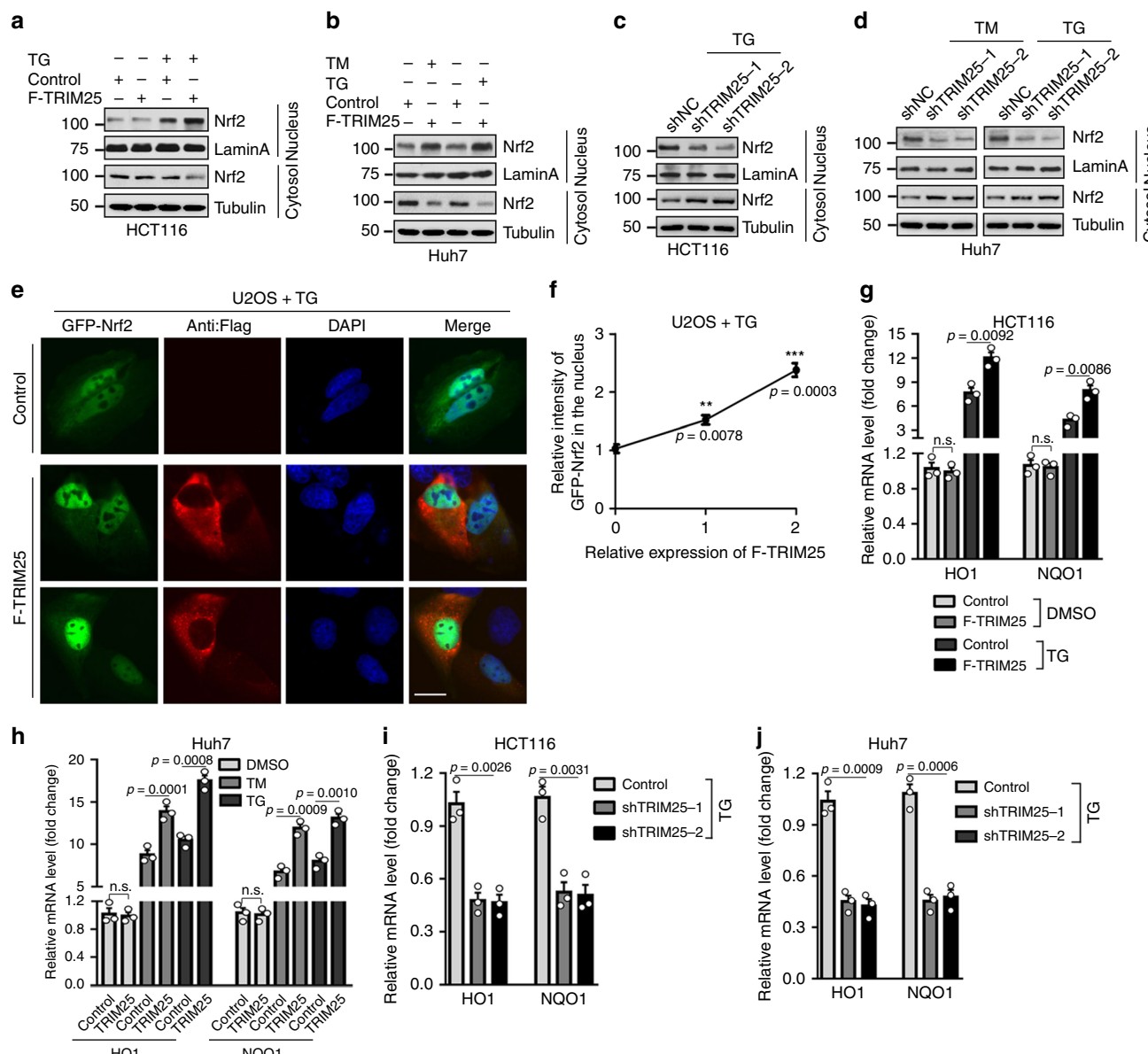

**Fig. 4 TRIM25 activates Nrf2 via promoting its nuclear import. a**, **b** Nuclear/cytosolic fractionation assay and western blot analysis of the level of Nrf2 in control and F-TRIM25-expressing HCT116 cells (**a**), or these cells were treated with TG (1 μM) for 12 h and the level of Nrf2 in the nuclear fraction was analyzed (**b**). LaminA is an internal control for nuclear fraction. **c** The level of Nrf2 in the nuclear fraction of HCT116 cells in control (shNC) or after stable knockdown of TRIM25, treated with TG (1 μM) for 12 h. **d** The level of Nrf2 in the nuclear fraction of Huh7 cells in control (shNC) or after stable knockdown of TRIM25, treated with TM (5 μg/ml) or TG (1 μM) for 12 h. **e**, **f** Localization (**e**) and quantification (**f**) of GFP-Nrf2 (green) in control or F-TRIM25 (red)-expressing U2OS cells, treated with TG (1 μM) for 6h. The nucleus is labelled by DAPI (blue). Scale bar, 10 μm. For quantification, 80 cells were for each group and the intensity of F-TRIM25 and GFP-Nrf2 in the nucleus was analyzed by Image J. And the relative intensity of GFP-Nrf2 in the nucleus of each group was relative to the control group. **g** Relative mRNA fold change of Nrf2 downstream genes: HO1, NQO1 in control and F-TRIM25-expressing HCT116 cells with stable knockdown of Nrf2, treated with or without TG (1 μM) for 6 h. **h** Relative mRNA fold change of Nrf2 downstream genes: HO1, NQO1 in control and F-TRIM25-expressing Huh7 cells with stable knockdown of Nrf2, treated with DMSO, TM (5 μg/ml) or TG (1 μM) for 12 h. **i**, **j** Relative mRNA fold change of Nrf2 downstream genes HO1 and NQO1 in HCT116 (**i**) and Huh7 (**j**) cells with control (shNC) or TRIM25 stably knockdown, treated with TG (1 μM) for 12 h. For **f**–**j**, data represent the mean ± SEM ($n = 3$). Statistical significance was assessed using two-tailed Student's t-tests. **$P < 0.01$, ***$P < 0.001$, n.s. not significant.

from TCGA database (Fig. S6a). These results demonstrate that TRIM25 promotes HCC tumor progression in vivo through Keap1/Nrf2 signaling.

**Upregulation of TRIM25 correlates with poor prognosis in HCC and multiple cancer types.** To reveal the clinical relevance of TRIM25 with cancers, TCGA pan-cancer interrogation of the mRNA transcript and genetic alterations of TRIM25 was

performed in human cancers. In contrast to normal adjacent tissues, TRIM25 mRNA levels were found notably upregulated in several cancerous tissues including liver cancer, breast cancer and low grade glioma (Fig. 7a and Supplementary Fig. 6b, c). Moreover, in HCC tissues expressed TRIM25 at significantly higher level relative to adjacent counterparts (Fig. 7b, c), suggesting the potential oncogenic activity of TRIM25 in HCC. We then examined TRIM25 expression in 90 cases of HCC surgical

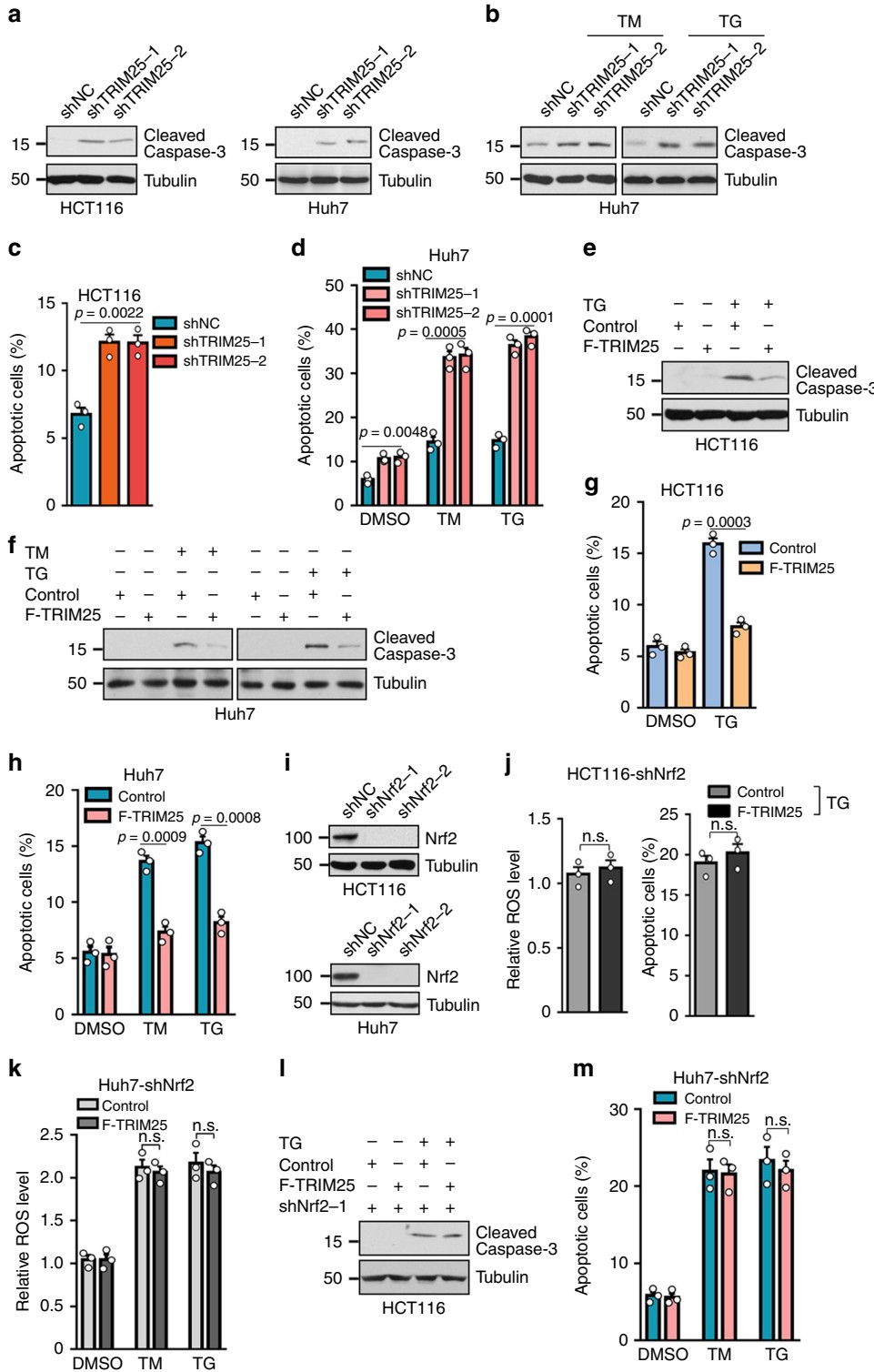

resected specimens, which were divided into two TRIM25 Low expression group (TRIM25[Low]) and TRIM25 High expression group (TRIM25[High]) (Fig. 7d). Survival analysis revealed that TRIM25[High] predicts reduced overall survival times than those with TRIM25[Low] (Fig. 7e). Consistently, data analysis from TCGA cohort also revealed higher TRIM25 expression in HCC patients was associated with poor clinical outcome as well as shorter disease-free survival times (Fig. 7f, g), which also observed in patients with breast cancer and lower grade glioma

(Supplementary Fig. 6d, e). Interestingly, HCC patients with TRIM25 amplification displayed shortened disease/progression-free survival and overall survival time compared on contrary to those without TRIM25 amplification (Fig. 7h, i). Taken together, these findings indicate that elevated levels of TRIM25 contribute to tumor progression, and serve as an important indicator for poor prognosis in several tumor types, especially HCC, thus providing a potential therapeutic target for cancer prevention.

**Fig. 5 TRIM25 negatively regulates tumor cells' ROS and apoptosis in cooperation with Nrf2. a** Western blot analysis of the levels of cleaved caspase3 in HCT116 and Huh7 cells for control (shNC) or after stable knockdown of TRIM25. **b, e** Western blot analysis of the levels of cleaved caspase3 in Huh7 cells for stalely knockdown (**b**) or overexpression (**e**) of TRIM25 treated with TM (5 μg/ml) or TG (1 μM) for 12 h. **c, g** Quantification of apoptotic HCT116 cells stable knockdown (**c**) or overexpression (**g**) of TRIM25, treated with or without TG (1 μM) for 12 hand analyzed by flow cytometry. **f** Western blot analysis of the levels of cleaved caspase3 in control and F-TRIM25-expressing HCT116 cells, treated with or without TG (1 μM) for 12 h. **d, h** Quantification of apoptotic Huh7 cells stable knockdown (**d**) or overexpression (**h**) of TRIM25, treated with TM (5 μg/ml) or TG (1 μM) for 12 h and analyzed by flow cytometry. **i** Western blot analysis of the efficiency of the stable cell knocks down of Nrf2 in HCT116 and Huh7 cells. **j** Quantification of ROS level and apoptotic HCT116 cells expressing control or F-TRIM25 and with simultaneously knockdown of Nrf2, treated with or without TG (1 μM) for 12 h. **k, m** Quantification of ROS level (**k**) and apoptotic (**m**) Huh7 cells expressing control or F-TRIM25 and with simultaneously knockdown of Nrf2, treated with TM (5 μg/ml) or TG (1 μM) for 12 h. **l** Western blot analysis of the levels of cleaved caspase3 in control and F-TRIM25-expressing HCT116 cells after stable knockdown of Nrf2, treated with or without TG (1 μM) for 12 h. For **c, d, g, h, j, k,** and **m,** data represent the mean ± SEM ($n = 3$). Statistical significance was assessed using two-tailed Student's $t$-tests. n.s. not significant.

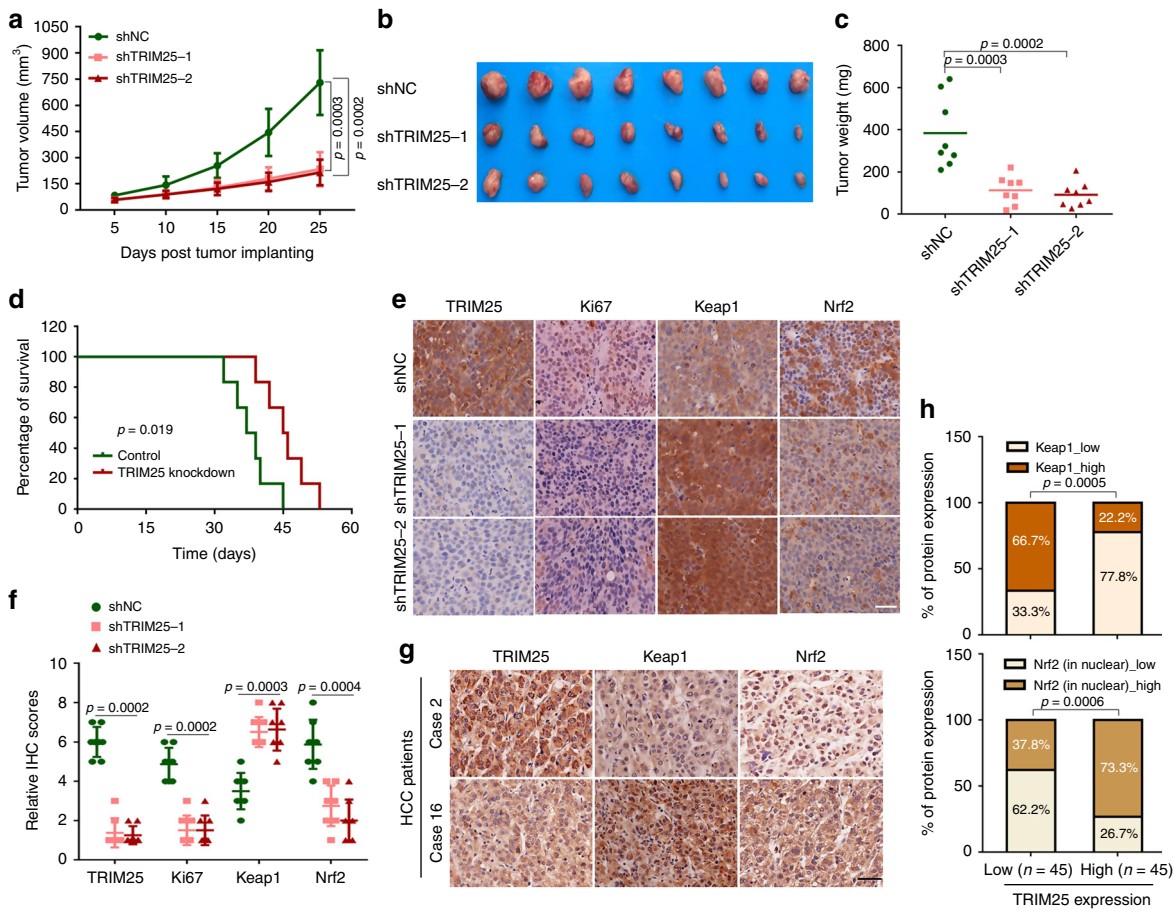

**Fig. 6 Knockdown of TRIM25 in HCC suppresses tumor growth in nude mice. a–c** Huh7 cells stably expressing the indicated shRNAs (shNC, shTRIM25-1 and shTRIM25-2) were subcutaneously injected in nude mice respectively. Shown are average tumor volumes over time ($n = 8$) (**a**) and representative image (**b**) and weights (**c**) of tumors at day 25. **d** Kaplan–Meier analysis of overall survival percentage in nude mice inoculated with Huh7-shNC or Huh7-shTRIM25. The statistical significance was assessed using two-sided log-rank test according to mice with depletion of control or TRIM25. **e, f** Representative images (**e**) of IHC staining and the relative IHC scores ($n = 8$) (**f**) of TRIM25, Ki67, Keap1 and Nrf2 in HCC tissues of mice inoculated with Huh7-shNC or Huh7-shTRIM25-1&-2. Scale bar, 50 μm. **g, h** Representative IHC staining images (**g**) and statistical data (**h**) of TRIM25, Keap1 and Nrf2 expression in HCC tissues (TRIM25 low expression group, $n = 45$ and TRIM25 high expression group, $n = 45$). Scale bar, 50 μm. For **a, c, f,** and **h,** data represent the mean ± SEM. Statistical significance was assessed using two-tailed Student's $t$-tests.

## Discussion

The ER is a major compartment that monitors the protein biosynthesis, assembly, and trafficking of secreted and membrane proteins. Cellular ER homeostasis is thus tightly controlled by the molecular machines involving ERAD and URP signaling[3]. Dysfunction of ER homeostasis, leading to the accumulation of misfolded proteins known as ER stress, is linked to many diseases including cancers[28]. Particularly, tumor cells are frequently exposed to microenvironmental disturbances that cause ER stress[1]. How tumor cells maintain ER homeostasis and survival remained not fully investigated. Moreover, TRIM proteins represent a large family encoded by human genome. Although they are extensively studied regarding their emerging roles in innate immunity[18,29], the

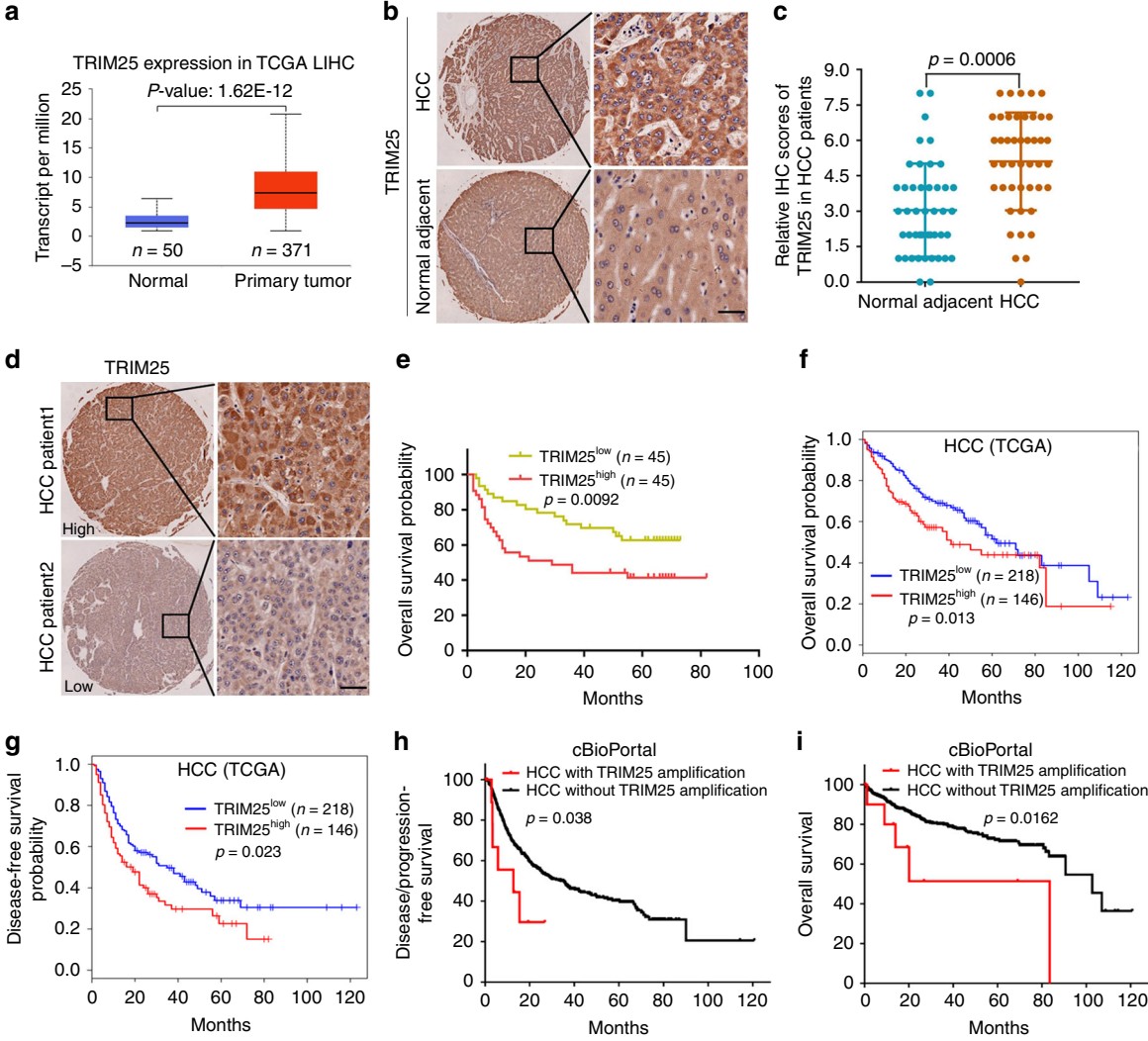

**Fig. 7 Upregulation of TRIM25 in HCC correlates with poor prognosis. a** Comparison of the TRIM25 mRNA level between HCC and normal tissues in TCGA database. Normal tissues/ Primary tumor tissues: $n = 50/371$, maximum $= 6.396/20.772$, upper quartile $= 3.461/10.888$, median $= 2.299/7.382$, lower quartile $= 1.608/4.781$, minimum $= 0.925/0.936$. Statistical significance was assessed using two-tailed Student's t-test. **b, c** Representative images (**b**) of IHC staining and the relative IHC scores (**c**) of TRIM25 in HCC tissues and adjacent normal tissues. Scale bar, 50 μm. Data represent the mean ± SEM (adjacent normal tissues, $n = 45$; HCC tissues, $n = 45$). The IHC score ranging from 0 to 8 was calculated by multiplying the staining extent score with the staining intensity score. Statistical significance was assessed using two-tailed Student's t-tests. **d, e** Representative images of IHC (**d**) and Kaplan–Meier analysis of overall survival probability (**e**) of TRIM25 levels in HCC patients. Scale bar, 50 μm. The statistical significance was assessed using two-sided log-rank test according to HCC patients with low or high expression of TRIM25(TRIM25 low expression patients, $n = 45$; TRIM25 high expression patients, $n = 45$). **f, g** The overall survival (**f**) and disease-free survival probability (**g**) were compared between TRIM25 High ($n = 146$) and Low expression ($n = 218$) in HCC patients from TCGA cohort. **h, i** Disease/progression-free survival (**h**) and overall survival (**i**) were compared between HCC with TRIM25 amplification ($n = 11$) and without TRIM25 amplification ($n = 594$) in HCC patients from cBioPortal database. For **e, f, g, h** and **i**, the statistical significance was assessed using two-sided log-rank test, log-rank p values were shown.

roles of TRIM family members in ER stress remains largely unknown. Here, by a systematic examination of TRIM proteins, we identified TRIM25 as a crucial regulator of ER stress that controls UPR signaling pathway and ERAD through Keap1/Nrf2 pathway, resulting in reduced ROS levels and ER stress induced apoptosis (Supplementary Fig. 6f).

TRIM25 likely directly ubiquitinates and degrades Keap1 through its ubiquitin E3 ubiquitin ligase, leading to the activation Keap1/Nrf2 pathway. This notion is supported by the failure of the ubiquitin ligase-defective mutant, TRIM25-2EA, to promote Keap1 ubiquitination and degradation. UPR signaling pathways can directly modulate Nrf2 through PERK-mediated phosphorylation[30]. Data gathered in our study suggested only a mild

activation of the PERK pathway was observed regardless of TRIM25 depletion or forced expression of TRIM25 upon ER stress in tumor cells, suggesting TRIM25 activates Nrf2 signaling that is independent of PERK pathway. Specifically, the IRE1-JNK signaling was found responsive to TRIM25 during ER stress, suggesting IRE1-JNK pathway is the downstream effector of TRIM25. It is not clear whether there is crosstalk between the IRE1-JNK pathway and the Keap1/Nrf2 pathway signaling, warranting further investigation in the future work. Here we show that TRIM25 is upregulated in response to ES stress. Moreover, overexpression or depletion of TRIM25 elicits a strong effect on Nrf2 activation, even though they only moderately affect the PERK signaling pathway. Thus, this upregulation of TRIM25

in response to ER stress likely provides a major mechanism that connects UPR with the Keap1-Nrf2 pathway. The mechanism of UPR-mediated activation of TRIM25 remains to be defined. We previously showed that certain TRIMs such as TRIM11 is upregulated by Nrf2[20]. If this is also the case for TRIM25, it would suggest that a positive feedback mechanism: a mild activation of Nrf2 leads to the upregulation of TRIM25, which in turn further stimulates Nrf2 activation via the degradation of Keap1. This would increase both the amplitude and duration of Nrf2 activation in response to oxidative stress.

The clinical relevance of TRIM25 in cancers including HCC has not been previously investigated. Liver cancer is the second leading cause of cancer-related death worldwide, resulting in ~800,000 fatalities annually[31]. Unlike most other cancers for which the mortality has declined, the incidence for liver cancer has been rising each year over the last 10 years in the US and worldwide, while the five-year survival remains at a dismal rate of ~18%[32,33]. The vast majority (~90%) of liver cancers are HCC. Although the risk factors for HCC are well known—including chronic infection of hepatitis B and C viruses and alcohol consumption, the molecular events driving the pathogenesis are incompletely understood[32,33]. The liver produces a large amount of secreted proteins, including major plasma proteins such as albumin and proteins involved in hemostasis and fibrinolysis, carrier proteins, hormones, prohormones, and apolipoprotein. HCCs are thought to raise from hepatocytes in the close proximity of terminal hepatic venule[34,35], which are especially active in producing secreted proteins. This, coupled with the rapid proliferation of HCCs, makes it likely that HCCs demand a highly robust capacity to maintain ER homeostasis. Moreover, Nrf2 is mutationally activated in 4–6% of HCCs[12,36,37], indicating the requirement for strong antioxidant and PQC systems in HCC. Here we demonstrate that the mRNA levels of TRIM25 are upregulated in a variety of cancer types including HCC and that upregulation of TRIM25 correlated with poor clinical outcome in patients with HCC as well as breast cancer and low grade glioma. We also find that TRIM25 expression is positively correlated with TP53 mutation in the human cancers including HCC and breast cancer (Supplementary Fig. 6g, h), implying that the increased expression of TRIM25 in cancer may be in part due to TP53 inactivating mutations. Supporting a role of TRIM25 in tumorigenesis, TRIM25 depletion in HCC cell line remarkably induces ER stress and impairs tumor cell growth in vitro and in vivo. Moreover, upregulation of TRIM25 is correlated with high Nrf2 expression and low Keap1 expression in both tumor xenograft and HCC specimens, suggested that TRIM25 upregulation may be in part responsible for Nrf2 activation in HCC and perhaps also other tumors that do not harbor mutations in Nrf2 or Keap1.

Our and other group previously demonstrated that TRIM11 as well as TRIM21 were implicated in modulating redox homeostasis through distinct mechanisms[20]. The present investigation unveiled that TRIM25 targets Keap1 for degradation, thereby Nrf2 activation, suggesting crosstalk between Keap1/Nrf2 signaling and TRIM family members functions as a common protecting mechanism to eliminate ROS production in response to ER stress. Therefore, it is of great interest to develop therapeutic approaches targeting TRIM family member(s) concurrently with Nrf2 inhibition for potential cancer interventions.

## Methods

**Plasmids and reagents.** Flag-TRIM25 (human) was a gift from Dong-Er Zhang (Addgene plasmid # 12449)[38], and CD3-δ-YFP was a.pngt from Nico Dantuma (Addgene plasmid # 11951)[24]. TRIM25 mutant (TRIM25-2EA) was also cloned into pcDNA3.1 vector with anNH2-terminal Flag tag. Lentiviral vectors expressing TRIM25 and TRIM25-2EA were constructed into pBabe-puro and pTRPE-GFP-T2A-mCherry (kindly provided by J. L. Riley, University of Pennsylvania) respectively. Lentiviral vectors expressing TRIM25 and Nrf2 shRNAs were purchased from Sigma: TRIM25 (TRCN0000272697and TRCN0000272699) and Nrf2 (TRCN0000273494 and TRCN0000284999). Reagents and its sources were indicated as followings: Tunicamycin (TM, Sigma) or Thapsigargin (TG, Sigma); Cycloheximide (CHX) (Calbiochem); 2′, 7′-dichlorodihydrofluoresceindiacetate (H2-DCFDA) (Sigma); complete protease inhibitor cocktail (Roche); protein A/G agarose (Thermo Fisher Scientific).

**Antibodies.** Primary antibodies against the following proteins were obtained from Cell Signaling Technology: BiP (#3183, 1:1000), JNK (#9252, 1:1000), Phospho-SAPK/JNK (Thr183/Tyr185) (#9251, 1:1000), eIF2α (#9722, 1:1000), Phospho-eIF2α (Ser51) (#9721, 1:1000), cleaved caspase-3 (Asp175) (#9661, 1:1000), Keap1 (P586) (#4678, 1:1000); from Sigma: Flag M2 (F3165, 1:2000), tubulin (T6074, 1:2000); from MBL: GFP (M048-3, 1:1000), His (D291-3, 1: 3000); from Thermo Fisher Scientific: HA (SG77) (71-5500, 1:1000); from Novus Biologicals: TRIM25 (NBP2-20710, 1:1000); from Santa Cruz Biotechnology: Nrf2 (A-10) (sc-365949, 1:1000); from Proteintech: KEAP1 (10503-2-AP, 1:200) and Nrf2 (16396-1-AP, 1:200); from Abclonal: TRIM25 (A12938, 1:200); from Cell Signaling Technology: anti-rabbit IgG, HRP-linked Antibody (7074, 1:3000) and anti-mouse IgG, HRP-linked Antibody (7076, 1:3000);from Invitrogen: Goat anti-Mouse IgG, Alexa Fluor 488 (A-11001, 1:500) and Donkey anti-Mouse IgG, Alexa Fluor 568 (A-10037, 1:500).

**Cell culture and stable cell lines.** HCT116 and U2OS cells were cultured in McCoy's 5A medium (Life Technologies), and Huh7, MCF7, MDA-MB-231 cells in DMEM (Life Technologies), both with 5% $CO_2$ at 37 °C and supplemented with 10% FBS (HyClone). MCF10A cells were cultured in DMEM/F12 supplemented with MEGM Single Quots TM Kit (Lonza).

**Quantitative real-time PCR.** Total RNA extraction of the cells was extracted using TRIzol (Invitrogen) and 1.5 μg RNA was reverse transcribed by the First Strand cDNA Synthesis Kit (Marligen Biosciences). Quantitative real-time (qRT) PCR was performed by SYBR Green PCR Master Mix (Applied Biosystems) in the ABI 7300 Detection System (Applied Biosystems), using the primers related to all TRIM genes described in the Supplementary Table 1[20,22]. The qRT-PCR primers related to sXBP1, ATF4, CHOP were showed in the Supplementary Table 1[39].

**Nuclear and cytoplasmic extraction.** The cells were suspended in buffer A (20 mM HEPES, 5 mM $CH_3COOK$, 1 mM $MgCl_2$, 0.5 mM DTT, pH 7.8) on ice for 15 min before they were broken with ~25 strokes of a glass Dounce homogenizer. The mixture was centrifuged at $1503 \times g$ for 5 min, and then the supernatant was aspirated and centrifuged at $20,000 \times g$ for 30 min to obtain the cytoplasmic fraction. To obtain the nuclear fraction, buffer B (20 mM HEPES, 5 mM $CH_3COOK$, 1 mM $MgCl2$, 0.5 mM DTT, 0.4MNaCl, pH 7.8) was used to suspend the pellet from the first centrifugation and this was centrifuged a second time at $18,407 \times g$ for 30 min.

**Immunoblotting and immunoprecipitation.** Cells were lysed in NP-40 lysis buffer (50 mM Tris-HCl, pH 8.8, 100 mM NaCl, 5 mM $MgCl_2$, 1 mM NaF, 0.5% NP-40, 2 mM DTT, 1 mM PMSF, and 1xcomplete protease inhibitor cocktail) for 30 min on ice and centrifuged at 4 °C at $16,000 \times g$ for 15 min to collect the supernatant. The protein concentrations were measured by Bradford assay (Bio-Rad Labs) and mixed with loading buffer before boiled. For the protein half-life analysis, cells were treated with CHX (50 μg/ml) for different times. These samples were resolved with SDS-PAGE and analyzed by western blot.

For Keap1 immunoprecipitation, cells were lysed in SDS-containing buffer, boiled, and diluted 20-fold in the above NP-40 lysis buffer. After centrifugation, the supernatant was incubated with Keap1 antibody at 4 °C overnight. Followed by washing with the NP-40 lysis 5 times, the beads were incubated with protein A/G agarose at 4 °C for 4 h. Finally, the beads were washed in the NP-40 lysis buffer five times, boiled and analyzed by western blot.

**MTT assay.** Two thousand five hundred cells per well were seeded in 96-well plates and cultured in DMEM medium. Viable cells were stained by 3-(4,5-dimethylthiazol-2-yl)-2,5-diphenyltetrazolium bromide (MTT) (Promega) and examined by measuring OD at 490 nm at the indicated time points.

**Apoptosis assay.** Cultured cells under different treatments were collected and analyzed using the FITC Apoptosis Detection Kit I (BD Biosciences) in order to detect apoptotic cells. Around $1 \times 10^6$ cells were suspended in $1 \times$ Binding Buffer and incubated with annexin V-FITC and propidium iodide (PI) at room temperature (25 °C) for 15 min. Then, the mixture was measured by flow cytometry and analyzed by Flow Jo software.

**Immunofluorescence.** Cells on the coverslips were washed 2 times in PBS, fixed with 4% PFA at 37 °C for 30 min and permeabilized with 0.10% Triton X-100 at room temperature for 20 min. Then, cells were washed 2 times in PBS and blocked with 3% BSA for 30 min before being incubated with the indicated primary and secondary antibodies at 4 °Covernight or at RT for 50 min, respectively. DAPI

(Vector Labs) was used to stain the nucleus and slides were observed using a fluorescence microscope (Olympus).

**Clinical samples and immunohistochemistry.** Tissue microarray of primary HCC samples were obtained from Shanghai Outdo Biotech Co., Ltd. and US Biomax Inc. (Rockville, MD, USA). Immunohistochemical (IHC) staining was performed as the following steps[40]. Formalin-fixed, paraffin-embedded tissue slides were dewaxed with xylene and rehydrated by a graded series of alcohols, followed by antigen retrieval and block with 5% BSA for 60 min. Incubation was carried out at 4 °C for overnight with the primary antibody. Primary antibodies included: anti-TRIM25 polyclonal antibody (1:200; Abclonal), anti-KEAP1 polyclonal antibody (1:200; Proteintech), and anti-Nrf2 polyclonal antibody (1:200; Proteintech). Signals were detected using Envision-plus detection system (Dako, Carpinteria, CA, USA) and visualized following incubation with 3,3′-diaminobenzidine.

**Tumor xenograft mouse models.** Tumor xenograft mouse models were established as the following steps[41,42]. Male athymic BALB/c nude mice (6 weeks old, Chinese Academy of Sciences) were raised in specific pathogen-free conditions, and were housed with a 12-h light/dark schedule at $25 \pm 1$ °C and were fed an auto-claved chow diet and water ad libitum. The mice were randomly divided into groups before injection. The tumor growth of Huh7-shNC and Huh7-shTRIM25-1&2 cells were determined following subcutaneous injection of cells into nude mice, respectively ($2.0 \times 10^6$ cell/mouse, eight mice/group). Four weeks post injection, the mice were sacrificed under anesthesia, and the tumor samples were then collected for further analysis. All animal experiments were undertaken in accordance with relevant guidelines and regulations and were approved by the Institutional Animal Care and Use Committee at SIAT.

**Statistics and reproducibility.** All experiments were repeated independently with similar results at least three times. The intensity of western blot bands and fluorescence signals were quantified using ImageJ (National Institutes of Health). Data analysis was obtained from GraphPad Prism 7 software (GraphPad Software, USA) through the unpaired two-tailed Student's t-test analysis.

**Reporting summary.** Further information on research design is available in the Nature Research Reporting Summary linked to this article.

## Data availability

The data supporting the findings in this study are available in the Article, Supplementary Information or from the corresponding author upon reasonable request.

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

## Acknowledgements

We thank Dr. Jianguo Chen and Junlin Teng at Peking University for critical suggestions and reading of the manuscript. This work was supported by the National Natural Science Foundation of China (31801186, 31701005, 81572832, and 81874174), Shenzhen Basic Science Research Project(JCYJ20170413153158716), SIAT Innovation Program for Excellent Young Researchers (201801), SIAT-GHMSC Biomedical Laboratory for Major Human Diseases, Shanghai Rising-Star Program (18QA1402600), Shanghai Municipal Commission of Health and Family Planning (2018YQ12) and School of Medicine, Shanghai Jiao Tong University (Excellent Youth Scholar Initiation Grant 17XJ11015 and 18XJ11006), as well as grants from National Institutes of Health (R01CA182675 and R01CA184867) and a sponsored research grant from Wealth Strategy Holding Limited to X.Y.

## Author contributions

Y.L., X.Y., X.W., and L.C. contributed to design of experiments, acquisition of data, interpretation of data and drafting the article. Y.L., S.T., LJ.L., Y.L., H.L., Z.L., LL.L., and L.C. contributed to acquisition of data. Y.L., X.Y., and L.C. contributed to analyze the data and write the manuscript.

## Competing interests

The authors declare no competing interests.
