## [Peer Review File · Nature Communications]

Reviewers' Comments:

Reviewer #1:

Remarks to the Author:

Liu et al report that a ubiquitin E3 ligase TRIM25 is induced upon ER stress and facilitates cell survival during ER stress by promoting Keap1 degradation thereby facilitating Nrf2-mediated antioxidant response. They also show that silencing TRIM25 leads to decrease xenograft tumor growth, and that TRIM25 expression correlates with poorer prognosis in hepatocellular carcinoma patients. The Keap1-Nrf2 antioxidant pathway is known to be regulated by Keap1 oxidation and autophagic degradation which release Nrf2 to migrate into the nucleus. The finding of an E3 ligase that directly interacts with and degrades Keap1 would be interesting and novel. The crosstalk between ER stress and redox regulation would also be an important topic. However, the data presented do not convincingly establish the TRIM25-Keap1 interaction and that Keap1 is directly ubiquitinated by TRIM25. Moreover, the underlying mechanism for the ER stress and redox crosstalk is not sufficiently explored. Therefore, while the biological and clinical relevance of TRIM25 in HCC are interesting, the main claims involving the mechanistic aspects are not convincingly established.

Major concerns

1. The authors use a relatively high dose of Thapsigargin (1 μ M) for the initial TRIM gene screening and subsequent experiments. It is recommended to discuss if TRIM25 induction at this dosage part of an adaptive response or an apoptotic response in the cell type used.
2. The authors state that TRIM25 suppresses IRE1/XBP1s signaling, which is the primary arm of the UPR known to activate ERAD genes (Lee et al Mol Cell Biol. 2003. 23:7448-7459). However, the authors provide data to suggest TRIM25 promotes ERAD activity which is conflicting to the above-mentioned literature.
3. The authors provide data for TG treatment at 12h to show no change in pEIF2 α (Fig. 1f). However, existing literature details that pEIF2 α is an early event in the ER stress response activated as early as 30 min post TG treatment. (Yan et al PNAS, 2002: 99: 15920–15925. Evaluation of a time course upto 12 hours will help understand the impact of TRIM25 over expression on the UPR response. In Fig. 1g, the text stated that TRIM25 overexpression down regulated phospho-eIF2 α , which does not seem to be the case from data provided in Fig. 1g.
4. Fig. 2 is to examine the connection between ER stress and ROS. It is not clearly stated or judging from the data what is the relationship between TRIM25 induced ROS and ER stress. The difference in ERAD measured by CD3-d-YFP shown in Fig. 2a-2e is quite marginal. Can the decreased ERAD be rescued by ROS scavengers and NRF2 activation?
5. Fig. 3 is critical to show TRIM25 binds to and ubiquitinate Keap1. However the data presented are quite weak. Fig. 3c shows only one-direction IP, and does not have the proper control of TRIM25 or Keap1 deficient cells. Does the co-IP work in other cell lines, and with other ER stressors? The decrease of Keap1 in Fig. 3d is marginal. Is Keap1 more stabilized in shTRIM25 cells? The result in Fig. 3f and 3g is marginal. It is hardly convincing that TRIM25 E3 activity promotes Keap1 degradation. It will be important to identify the lysine residues on Keap1 that are ubiquitinated by TRIM25, which will help nail down the mechanism.
6. Fig. 4 needs to show cytosolic fractions. What is the size of the Nrf2 bands?
7. The authors claim decreased cell proliferation in TRIM25 knockdown samples (line 197) but is actually a decrease in cell viability as opposed to proliferation. In line 204, the authors claim that knockdown of TRIM25 elicits no effect on cell proliferation but data in S4a suggests otherwise.
8. The authors discuss that PERK pathway has previously been reported to directly activate Nrf2 and provide evidence for no activation of pEIF2 α at the 12h timepoint. The original article by Cullinan et al MCB (2003) 7198-7209, states that PERK kinase directly phosphorylates Nrf2 and regulates its nuclear translocation upto 6h post TG treatment. A time course study of these molecules will help delineate the sequence of events and the contribution of PERK/Nrf2 and IRE1/Keap1-Nrf2 pathways to activate the antioxidant response.

Minor :

1. Mention the timeline for experiments in 1i, 4i, 4j,5a
3. Fig. 3c with Anti-IgG and Anti-TRIM25 top of the gel then with IP: Keap1 on the side.
4. Check for grammatical and typographical errors

Reviewer #2:

Remarks to the Author:

1. In Figure 3c, the authors showed that the interaction between TRIM25 and KEAP1 is increased under ER stress. Is it due to increase in TRIM25 abundance, or quality change of TRIM25-Keap1 interaction? As protein levels of TRIM25 and KEAP1 must be changed by the TG treatment, input protein of KEAP1 and TRIM25 should be shown in Figure 3c.
2. To give an insight into how TRIM25 regulates KEAP1, interacting domains of both TRIM25 and KEAP1 should be determined. This is an important information to understand the interaction.
3. As KEAP1 is reported to degrade through the selective autophagy (Taguchi et al, 2016), it seems necessary to examine whether the degradation of KEAP1 is mediated by the proteasome pathway or autophagy pathway.
4. Increased expression of TRIM25 by ER stress and in cancer is one of the key findings of this study. The authors should discuss mechanisms how TRIM25 expression is regulated.

Minor points

In Figure 3c, is this coIPed with anti-TRIM25? "IP: KEAP1" may be mis-labelling?

Response to reviewers:

We thank the reviewers for their interests in our work and their insightful and constructive comments and criticisms. Accordingly, we have performed a large number of new experiments and extensively revised the manuscript. We have also expanded the Discussion sections. Please find below our responses to all comments and criticisms. The data generated during the revision are highlighted in red color in the figures and the text.

Reviewer #1 (Remarks to the Author):

Liu et al report that a ubiquitin E3 ligase TRIM25 is induced upon ER stress and facilitates cell survival during ER stress by promoting Keap1 degradation thereby facilitating Nrf2-mediated antioxidant response. They also show that silencing TRIM25 leads to decrease xenograft tumor growth, and that TRIM25 expression correlates with poorer prognosis in hepatocellular carcinoma patients. The Keap1-Nrf2 antioxidant pathway is known to be regulated by Keap1 oxidation and autophagic degradation which release Nrf2 to migrate into the nucleus. The finding of an E3 ligase that directly interacts with and degrades Keap1 would be interesting and novel. The crosstalk between ER stress and redox regulation would also be an important topic. However, the data presented do not convincingly establish the TRIM25-Keap1 interaction and that Keap1 is directly ubiquitinated by TRIM25. Moreover, the underlying mechanism for the ER stress and redox crosstalk is not sufficiently explored. Therefore, while the biological and clinical relevance of TRIM25 in HCC are interesting, the main claims involving the mechanistic aspects are not convincingly established.

Major concerns

1. *The authors use a relatively high dose of Thapsigargin (1 μ M) for the initial TRIM gene screening and subsequent experiments. It is recommended to discuss if TRIM25 induction at this dosage part of an adaptive response or an apoptotic response in the cell type used.*

Response: We thank the reviewer for this suggestion. To address the reviewer's concern, we performed an MTT assay to examine the survival of HCT116 and Huh7 cells that were treated with Thapsigargin (TG, 1 μ M) for 12h, a condition that used in our initial survey of TRIM genes. We found that this treatment did not significantly change cell viability (please see the figure shown below), suggesting that the induction of TRIM25 at this dosage is not an apoptotic response. This is consistent with the role of TRIM25 in protecting against ER stress, as shown in the current manuscript.

2. The authors state that TRIM25 suppresses IRE1/XBP1s signaling, which is the primary arm of the UPR known to activate ERAD genes (Lee et al Mol Cell Biol. 2003. 23:7448-7459). However, the authors provide data to suggest TRIM25 promotes ERAD activity which is conflicting to the above-mentioned literature.

Response: We thank the reviewer for raising this important issue. We find that TRIM25 promotes redox balance through the activation of Nrf2. This likely facilitates protein folding in the ER compartment, thereby attenuating UPR when cells are treated with ER stressors. The effect of TRIM25 on ERAD may also be related to redox balance. For example, ROS can damage proteins involved in the recognition, retro-translocation, and degradation of ERAD substrates including, for example, the ubiquitin E3 ligases for ERAD substrates including HRD1 and DOA10 as well as the proteasome. Thus, by reducing ROS levels via the activation of Nrf2, TRIM25 likely promotes the degradation of ERAD substrate, even though it can reduce UPR response. This may in turn contribute to the attenuated UPR response in TRIM25-overexpressing cells.

3. The authors provide data for TG treatment at 12h to show no change in pEIF2 α (Fig. 1f). However, existing literature details that pEIF2 α is an early event in the ER stress response activated as early as 30 min post TG treatment. (Yan et al PNAS, 2002: 99: 15920–15925. Evaluation of a time course up to 12 hours will help understand the impact of TRIM25 over expression on the UPR response. In Fig. 1g, the text stated that TRIM25 overexpression down regulated phospho-eIF2 α , which does not seem to be the case from data provided in Fig. 1g.

Response: We appreciate the reviewer mentioning this interesting data. According to our results, we claim that the impact of TRIM25 on the UPR response was mainly on phospho-JNK, not phospho-eIF2 α . Actually, knockdown of TRIM25 slightly increase the protein level of phospho-eIF2 α (Fig. 1f). Upon TG treatment, phospho-eIF2 α was activated and overexpression of TRIM25 slightly decrease the protein level phospho-eIF2 α (Fig. 1g, lighter exposure of phospho-eIF2 α was showed now). Meanwhile, TRIM25 has an obvious effect on negative regulating the phospho-JNK upon these conditions (Fig. 1f, g). Together, these data proving that the effect of TRIM25 on the UPR was mainly through the JNK signaling pathway.

4. Fig. 2 is to examine the connection between ER stress and ROS. It is not clearly stated or judging from the data what is the relationship between TRIM25 induced ROS and ER stress. The difference in ERAD measured by CD3-d-YFP shown in Fig. 2a-2e is quite marginal. Can the decreased ERAD be rescued by ROS scavengers and NRF2 activation?

Response: To address these comments, we have repeated these experiments and got more representative blots and obvious statistic results (Fig. 2a-2e). Furthermore, we used ROS scavengers NAC and NRF2 activator tBHQ to treated TRIM25 depleted cells and found the decreased ERAD can be partially rescued by these two drugs (Supplementary Fig. 2h), indicating that TRIM25 positively regulating ERAD by partially modulating Nrf2 activation

and the cellular ROS levels.

5. Fig. 3 is critical to show TRIM25 binds to and ubiquitinate Keap1. However the data presented are quite weak. Fig. 3c shows only one-direction IP, and does not have the proper control of TRIM25 or Keap1 deficient cells. Does the co-IP work in other cell lines, and with other ER stressors? The decrease of Keap1 in Fig. 3d is marginal. Is Keap1 more stabilized in shTRIM25 cells? The result in Fig. 3f and 3g is marginal. It is hardly convincing that TRIM25 E3 activity promotes Keap1 degradation. It will be important to identify the lysine residues on Keap1 that are ubiquitinated by TRIM25, which will help nail down the mechanism.

Response: To address these comments, we have performed several experiments:

- (1) We have now included both the input and control IgG IP as controls for the TRIM25-Keap1 co-IP experiment (Fig. 3c).
- (2) We have now added the co-IP results using another cell line, Huh7, to confirm the TRIM25-Keap1 interaction (Fig. 3d). Moreover, this interaction was increased in cells treated with either TG (Fig. 3d) or the other ER stressor, TM (5 μ g/ml) (Supplementary Fig. 3a).
- (3) We have now showed the other-direction IP, GST-Keap1 IP TRIM25 (Fig. 3e). This in vitro experiment using purified recombinant proteins also suggest a direct interaction between TRIM25 and Keap1.
- (4) We have repeated the previous experiments of Fig. 3d and observed a stronger and more consistent effect of TRIM25 on Keap1 (Fig. 3h).
- (5) We determined the half-life of Keap1 and observed that knockdown of TRIM25 using two independent shRNAs results in a strong stabilization of Keap1 (Fig. 3f, g).
- (6) We have repeated the previous experiments of Fig. 3f and found that TRIM25, but not the ubiquitin-defective mutant TRIM25-2EA (Fig. 3m), strongly promotes Keap1 degradation.
- (7) We have now identified the lysine residues (K615) on Keap1 that are ubiquitinated by TRIM25 (Fig. 3r).

6. Fig. 4 needs to show cytosolic fractions. What is the size of the Nrf2 bands?

Response: We have now showed the cytosolic fractions and provided the molecular weight marker (Fig. 4a-4d).

7. The authors claim decreased cell proliferation in TRIM25 knockdown samples (line 197) but is actually a decrease in cell viability as opposed to proliferation. In line 204, the authors claim that knockdown of TRIM25 elicits no effect on cell proliferation but data in S4a suggests otherwise.

Response: We apologize for the confusion. Our data show that while TRIM25 knockdown decreased cell viability in the absence of ER stress, force expression of TIRM25 did not elicit

apparent effect on cell viability unless cells are treated with ER stressor. We have now revised the text accordingly (line 220-229).

8. *The authors discuss that PERK pathway has previously been reported to directly activate Nrf2 and provide evidence for no activation of pEIF2 α at the 12h time point. The original article by Cullinan et al MCB (2003) 7198-7209, states that PERK kinase directly phosphorylates Nrf2 and regulates its nuclear translocation up to 6h post TG treatment. A time course study of these molecules will help delineate the sequence of events and the contribution of PERK/Nrf2 and IRE1/Keap1-Nrf2 pathways to activate the antioxidant response.*

Response: We observed pEIF2 α at the 12h after stimulation with ER stressors (Fig. 1g). We have also performed a time course experiment, which indicate that Nrf2 is activated moderately at 6 h and strongly at 12 h post TG treatment (please see the figure shown below). Our previous results suggested that TRIM25 depletion or overexpression only mildly affects the PERK pathway, suggesting TRIM25 activates Nrf2 signaling at least in part through Keap1.

Minor:

1. *Mention the timeline for experiments in 1i, 4i, 4j, 5a*

Response: We have now mentioned the timeline for these experiments.

2. *Fig. 3c with Anti-IgG and Anti-TRIM25 top of the gel then with IP: Keap1 on the side.*

Response: We apologize for this confusing labelling, and have now corrected them.

3. *Check for grammatical and typographical errors*

Response: We thank the reviewer for pointing out these grammatical and typos errors in the manuscript, and have fixed grammatical and typographical errors. We have also extensively revised all sections of the manuscript, especially the main text, to make them more concise and clearer and to provide appropriate background and context for our study.

Reviewer #2 (Remarks to the Author):

1. In Figure 3c, the authors showed that the interaction between TRIM25 and KEAP1 is increased under ER stress. Is it due to increase in TRIM25 abundance, or quality change of TRIM25-Keap1 interaction? As protein levels of TRIM25 and KEAP1 must be changed by the TG treatment, input protein of KEAP1 and TRIM25 should be shown in Figure 3c.

Response: We have now showed the Keap1 and TRIM25 levels in the cell lysates (Fig. 3c, d). As expected, TRIM25 levels are increased and Keap1 levels are decreased upon TG treatment. However, we adjusted the amount of immunoprecipitates so that the untreated and treated samples comparable amounts of TRIM25. Under this condition, despite the reduction of Keap1 in the lysates, more Keap1 protein is associated with TRIM25 (Fig. 3c, d). Therefore, the increased interaction is likely due to an increase in the affinity between TRIM25 and Keap1.

2. To give an insight into how TRIM25 regulates KEAP1, interacting domains of both TRIM25 and KEAP1 should be determined. This is an important information to understand the interaction.

Response: We appreciate this insightful comment from the reviewer and have now determined the interacting domains of both TRIM25 and KEAP1, which showing that N-terminal of TRIM25 containing RING domain and double glycine repeat (DGR) domain of Keap1 are crucial for their interaction (Fig. 3j, k, p, q). We have also determined the Lys residue (K615) on Keap1 that is important for TRIM25-mediated ubiquitination.

3. As KEAP1 is reported to degrade through the selective autophagy (Taguchi et al, 2016), it seems necessary to examine whether the degradation of KEAP1 is mediated by the proteasome pathway or autophagy pathway.

Response: By using the proteasome inhibitor MG132 and autophagy inhibitor CQ, we have now found that the degradation of KEAP1 is mediated by the proteasome pathway, rather than autophagy (Fig. 3i).

4. Increased expression of TRIM25 by ER stress and in cancer is one of the key findings of this study. The authors should discuss mechanisms how TRIM25 expression is regulated.

Response: Using the bioinformatics analysis, we found that TRIM25 expression had positively correlated with TP53 mutation in the human cancers including liver cancer and breast cancer (please see the figure shown below), implying that increased expression of TRIM25 in cancer may be due to TP53 inactivating mutations. Moreover, we previously showed that certain TRIMs such as TRIM11 is upregulated by Nrf2. If this is also the case for TRIM25, it would suggest that a positive feedback mechanism: a mild activation of Nrf2 leads to the upregulation of TRIM25, which in turn further stimulates Nrf2 activation via the degradation of Keap1. Thus, an alternative, but not mutually exclusive, scenario is that tumor cells, which commonly experience redox stress, upregulate Nrf2 leading to the positive

feedback through TRIM25 to further bolster the expression of both Nrf2 and TRIM25. We have now discussed this point in the revised manuscript.

Minor points

In Figure 3c, is this coIPed with anti-TRIM25? "IP: KEAP1" may be mis-labelling?

Response: This co-IP uses anti-TRIM25 antibody. We have now corrected this error (Fig. 3c).

Reviewers' Comments:

Reviewer #1:

Remarks to the Author:

The authors have done substantial amount of work to address previously raised concerns. The paper is much improved and is ready to publish.

Reviewer #2:

Remarks to the Author:

This manuscript has been nicely improved through the revise.

The following is a point-to-point response to the reviewers' comments.

REVIEWERS' COMMENTS:

Reviewer #1 (Remarks to the Author):

The authors have done substantial amount of work to address previously raised concerns.

The paper is much improved and is ready to publish.

Response:

We are grateful to this reviewer for taking the time to read our manuscript and give us very helpful comments.

Reviewer #2 (Remarks to the Author):

This manuscript has been nicely improved through the revise.

Response:

We are grateful to this reviewer for taking the time to read our manuscript and give us very helpful comments.